# Can Ampullary G1 and G2 Neuroendocrine Tumors Be Cured by Endoscopic Papillectomy?

**DOI:** 10.3390/jcm12062286

**Published:** 2023-03-15

**Authors:** Wengang Zhang, Ningli Chai, Enqiang Linghu

**Affiliations:** Department of Gastroenterology, The First Medical Center of Chinese PLA General Hospital, Beijing 100853, China; zhangwengangcr@163.com (W.Z.); chainingli@vip.163.com (N.C.)

**Keywords:** ampullary neuroendocrine tumors, endoscopic papillectomy

## Abstract

Objectives: At present, pancreaticoduodenectomy or local excision are the main treatment options for ampullary neuroendocrine tumors of any size with no distant spread. Endoscopic papillectomy provided a super minimally invasive treatment method for ampullary lesions. However, the studies of endoscopic papillectomy for ampullary neuroendocrine tumors were very limited. This study aimed to assess the feasibility of endoscopic papillectomy for ampullary neuroendocrine tumors. Methods: Between August 2007 and June 2021, seven patients with ampullary neuroendocrine tumors with no advanced signs underwent endoscopic papillectomy in our center. We assessed and analyzed the related clinical outcomes. Moreover, a comprehensive literature review was conducted. Results: All the seven cases underwent endoscopic papillectomy successfully; six (85.7%) of them achieved the complete resection. No recurrence occurred over a median follow-up of 48 months (range 6–172 months). Moreover, 22 cases from the comprehensive search were included, and showed a promising clinical outcome. Conclusions: Endoscopic papillectomy appeared to be a feasible way to treat ampullary neuroendocrine tumors without the advanced signs, although further prospectively multicenter studies are warranted.

## 1. Introduction

A neuroendocrine tumor (NET) is defined as an epithelial neoplasm that shows neuroendocrine differentiation when analyzed by conventional histological, immunohistochemical, ultrastructural and biological evaluation [1]. Ampullary neuroendocrine tumors are extremely rare tumors, accounting for less than 2% of all tumors of the ampullary region [2]. Owing to the special location at the ampulla, ampullary neuroendocrine tumors mostly present with obstructive jaundice, non-specific upper abdominal pain, pancreatitis or weight loss. Different from other gastrointestinal lesions, the endoscopic removal of papillary lesions is far more difficult and complex for several reasons, including its complicated location in the duodenum, the extremely enriched vasculature, and the thin wall of the duodenum. Therefore, pancreaticoduodenectomy or local excision is recommended for the treatment of ampullary neuroendocrine tumors of any size with no distant spread, although this treatment will cause relatively large trauma [3]. Recently, endoscopic papillectomy has become a well-established treatment for patients with ampullary adenoma, given the lesser trauma [4]. However, to date, the studies of endoscopic papillectomy for ampullary neuroendocrine tumors were very limited and only some sporadic case reports were published [5,6,7]. Therefore, the present study aimed to assess the feasibility of endoscopic papillectomy for ampullary neuroendocrine tumors (G1 or G2) based on the data from our center and a comprehensive literature review.

## 2. Methods

This retrospective study was approved by the Ethics Committee of the Chinese People’s Liberation Army General Hospital. Informed consent was obtained for all patients described. Between August 2007 to June 2021, seven patients with G1 or G2 ampullary neuroendocrine tumors successfully underwent endoscopic papillectomy at this hospital. The pathologic sections of these seven cases were diagnosed again according to the WHO classification of digestive system tumors (5th Edition, 2019). Of note, the procedures of endoscopic papillectomy for ampullary mixed lesions with NET and adenoma/adenocarcinoma were excluded. Before the procedures, computerized tomography, magnetic resonance imaging and endoscopic ultrasonography were performed to rule out the distant and regional lymph nodes metastasis; and endoscopic retrograde cholangiopancreatography (ERCP) was performed to ensure that there was no tumor invasion within the bile or pancreatic ducts. The procedures of endoscopic papillectomy were performed as follows (Figure 1): (1) The patient was placed in a supine position under general anesthesia; (2) Duodenoscopy (TJF-240/TJF-260V; Olympus, Tokyo, Japan) was used to observe the papillary lesion; (3) A submucosal injection was performed to lift the lesion, or an injection was not performed; (4) En bloc resection of the lesion was performed with an endoloop (SD-7P-1/SD-221L-25; Olympus, Tokyo, Japan); (5) A guidewire was used to select the bile duct (BD) and/or the pancreatic duct (PD); (6) Electric coagulation forceps and argon plasma coagulation were used for wound hemostasis, and then the wound was closed with metal clips if necessary; (7) A BD and/or PD stent was inserted; (8) Fibrin glue (S201110020; Bioseal, Guangzhou, China) was sprayed on the closed wound if necessary; (9) Exhaust air in the stomach and stent placement were checked under X-ray; and (10) The sample was sent for pathology. Of note, no other further supporting treatments, such as interferon alpha or molecular treatment, were performed for these 7 patients.

Patients in study were observed closely after endoscopic papillectomy treatment for complications, including postoperative pancreatitis, perforation, bleeding and abdominal infection. Corresponding treatment was given once complications were encountered. Patients were kept fasting for five days after the procedure, and a liquid diet was followed for an additional day if no complications occurred. Diet was gradually restored to normal from the seventh day. Patients would be discharged on the eighth day if there were no complications observed, otherwise the hospital discharge would be delayed at the discretion of the endoscopists. Postoperative medications mainly included proton pump inhibitor (PPI), somatostatin and antibiotics. PPI, including esomeprazole (20 mg, twice a day), rabeprazole (20 mg, once a day) and so on, was required for at least 2 weeks.

The patients were followed up with endoscopy and abdominal magnetic resonance imaging or computerized tomography (plain plus contrast-enhanced scanning) at three, six months, and yearly after endoscopic papillectomy to assess the recrudesce of ampullary neuroendocrine tumors.

Complete resection (R0) was defined as an “en bloc” resection without lateral or vertical margin involvement and without lymphovascular invasion on the resected specimen. Hemorrhages were defined as either a drop in hemoglobin ≥2 g/dL or a clinically overt bleed.

Moreover, we also conducted a comprehensive search on the Pubmed database. The search was performed using the following strategy: ampullary neuroendocrine tumor OR ampullary neuroendocrine neoplasm OR neuroendocrine tumor of the ampulla of Vater OR ampullary carcinoid OR gastroduodenal neuroendocrine tumor OR gastroduodenal neuroendocrine neoplasm. The last search date was December 2021. We selected the papers in which endoscopic papillectomy was performed for ampullary neuroendocrine tumors. Of note, those cases of endoscopic papillectomy for ampullary mixed lesions with NET and adenoma/adenocarcinoma were not included; the cases of endoscopic mucosal resection (EMR) for periampullary neuroendocrine tumor were not included; the cases of NET of minor papilla were not included; a case of endoscopic submucosal dissection (ESD) for ampullary neuroendocrine tumor was not included. Moreover, we did not include the studies in which the main topic was endoscopic papillectomy for ampullary adenomas/adenocarcinomas, and only some sporadic ampullary neuroendocrine tumors cases without the detailed information were reported.

Statistical analysis: Nonparametric data are expressed as medians.

## 3. Results

For the seven procedures in our center, endoscopic papillectomy was successfully performed in all patients, and the clinical characteristics and outcomes were shown in Table 1. The median tumor size was 15 mm (range 6–30 mm). Postoperative pathological diagnosis revealed four G2 NETs and three G1 NETs. Complete resection was achieved in six cases and one case was confirmed with lymph-vascular invasion postoperatively; this case refused additional treatment and there was no recurrence after a follow-up of 30 months. In terms of adverse events, postoperative hemorrhages occurred in one case, and successful hemostasis was achieved by metal clips under endoscopy. As for endoscopic morphology, four G2 cases shown a performance of surface depression (Figure 1i) and three G1 cases shown a performance of smooth surface with pannus (Figure 1d). After a median 48 months follow-up (range 6–172 months), no recurrence was encountered.

In terms of the results of the literature review, a total of 22 cases were included; of these, 12 cases were described detailed in the form of case reports and the clinical characteristics and outcomes were shown in Table 2 [5,6,7,8,9,10,11,12,13,14,15]. Of these 12 cases, three did not report the follow-up results, and no recurrence occurred during a median 16 months (range 6–96 months) for the other nine cases. Gincul et al. [16] published a study focusing on the endoscopic treatment for small duodenal and ampullary neuroendocrine tumors, and in which seven cases with ampullary neuroendocrine tumors underwent endoscopic papillectomy; one case was lost follow-up; five cases achieved complete resection; and no recurrence occurred. Galetti et al. [17] published a study in which endoscopic papillectomy was performed for three neuroendocrine tumors of major papilla, although the detailed follow-up information was not provided.

## 4. Discussion

Ampullary neuroendocrine tumors are an extremely uncommon subset of pancreatic cancer that have a distinct clinical and morphological profile. At present, strategic therapeutic methods have not yet been established for ampullary neuroendocrine tumors. Pancreaticoduodenectomy and local excision are the main options for ampullary neuroendocrine tumors. The drawbacks of these surgical techniques are obvious and include the high adverse event rate and mortality, long anesthesia duration, large operative trauma, high cost, etc. Compared with conventional surgical techniques for ampullary lesions, endoscopic papillectomy has the merit of less trauma that has popularized its rapid worldwide use. However, to date, the studies focusing on the treatment outcomes of endoscopic papillectomy for ampullary neuroendocrine tumors were very limited. The present study confirmed that endoscopic papillectomy, which enables complete removal for the ampullary tumor with less trauma, achieved a promising outcome for G1 and G2 ampullary neuroendocrine tumors. The main advantage of endoscopic papillectomy over surgical techniques lies in the reserved anatomical structure of the digestive system of patients, which could improve the postoperative quality of life significantly. Moreover, the results of the literature review further verified the satisfactory outcomes of endoscopic papillectomy for ampullary neuroendocrine tumors. Of note, the cases from our center consisted of larger-size tumors and more G2 cases than the cases already reported [5,6,7,8,9,10,11,12,13,14,15,16,17].

There is still one matter for concern, the preoperative computerized tomography, magnetic resonance imaging, endoscopic ultrasonography and ERCP examinations are necessary to rule out the distant metastasis, regional lymph nodes metastasis and tumor invasion within the bile or pancreatic duct. That is, endoscopic papillectomy could be performed for ampullary neuroendocrine tumors only when distant metastasis, regional lymph nodes metastasis and tumor invasion within the bile or pancreatic duct were eliminated, otherwise surgical techniques or other treatments should be considered.

The major limitation of our study was the nature of retrospective study from a single center. However, to the best of our knowledge, this paper represented the largest sample capacity of endoscopic papillectomy for G1 and G2 ampullary neuroendocrine tumors up to now, and more detailed clinical characteristics and outcomes were provided. Moreover, we provided the cases with larger tumor size and longer follow-up.

Endoscopic papillectomy might be a feasible and effective way to treat G1 and G2 ampullary neuroendocrine tumors without distant metastasis, regional lymph nodes metastasis and the bile/pancreatic duct invasion. Moreover, further prospectively multicenter studies are warranted.

## Figures and Tables

**Figure 1 jcm-12-02286-f001:**
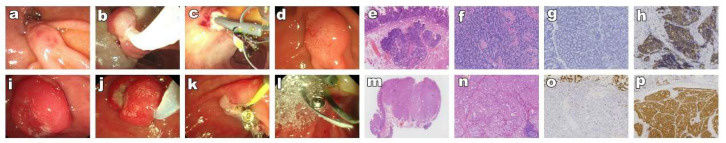
The typical cases of endoscopic papillectomy for ampullary neuroendocrine tumors. (**a**) The endoscopic morphology of case 5 (Table 1). (**b**) The tumor was removed by an endoloop (Case 5, Table 1). (**c**) The wound was closed with metal clips and the stent of bile duct was inserted (Case 5, Table 1). (**d**) The typical endoscopic morphology of pannus of G1 ampullary neuroendocrine tumors (Case 3, Table 1). (**e**) Low power histological view of the resected specimen (Case 5, Table 1). (**f**) The tumor cell presented an arrangement of strip shape (Case 5, Table 1). (**g**) The tumor cells showed positivity for Ki-67 (1%) (Case 5, Table 1). (**h**) Positive synaptophysin staining (Case 5, Table 1). (**i**) The endoscopic morphology of case 2 with surface depression (Table 1). (**j**) The tumor was removed by an endoloop after submucosal injection (Case 2, Table 1). (**k**) The wound was closed with metal clips and the guide wire was inserted into the bile duct (Case 2, Table 1). (**l**) The stent of bile duct was inserted (Case 2, Table 1). (**m**) Low power histological view of the resected specimen (Case 2, Table 1). (**n**) The tumor cell presented an arrangement of adenoid shape. (**o**) The tumor cells showed positivity for Ki-67 (3%) (Case 2, Table 1). (**p**) Positive synaptophysin staining (Case 2, Table 1).

**Table 1 jcm-12-02286-t001:** Clinical characteristics and outcomes for seven patients with ampullary neuroendocrine tumors after endoscopic papillectomy (from our center).

Case no.	Age (y)/Sex	Clinical Symptoms (%)	Endoscopic Morphology	Tumor Size (mm) *	Operation Time (min)	Submucosal Injection	Pathologic Diagnosis	Complete Resection (R0)	Placement of Stent	Postoperative Adverse Events	Recurrence	Follow-Up (Month)
1	32/male	None	Surface depression	30	43	No	G2	Yes	Sheet of PD	Postoperative hemorrhages	No	172
2	67/ male	None	Smooth surface with pannus	15	19	No	G1	Yes	No	None	No	124
3	78/female	None	Smooth surface with pannus	6	31	No	G1	Yes	No	None	No	53
4	72/ male	Intermittent abdominal pain	Surface depression	20	17	No	G2	Yes	Sheet of BD	None	No	48
5	51/ male	None	Smooth surface with pannus	6	21	Yes	G1	No	Sheet of BD	None	No	30
6	56/ male	None	Surface depression	20	90	No	G2	Yes	Sheet of BD	None	No	25
7	54/ female	None	Surface depression with pannus	15	16	No	G2	Yes	Sheet of BD	None	No	6

BD, bile duct. PD, pancreatic duct. *, the tumor size was measured by pathologic tissue section.

**Table 2 jcm-12-02286-t002:** Clinical characteristics and outcomes for patients with ampullary neuroendocrine tumors after endoscopic papillectomy (from published studies).

No.	Reference	Number of Case	Age (y)/Sex	Clinical Symptoms (%)	Endoscopic Morphology	Tumor Size (mm)	Submucosal Injection	Pathologic Diagnosis	Complete Resection (R0)	Placement of Stent	Postoperative Adverse Events	No Recurrence (Months)
1	5	1	51/male	Repeated back pain	Smooth surface with pannus	9.4 (EUS)	No	G1	ND	Sheet of PD	None	96
2	5	1	77/male	None	Surface depression	9.6 (EUS)	No	G1	ND	ND	None	32
3	6	1	62/ male	None	Surface depression	7.0 (EUS+ Pathologic analysis)	Yes	Carcinoid	Yes	Sheet of PD and BD	None	6
4	7	1	71/ male	melena	Surface depression	15.0 (Endoscopy)	Yes	Carcinoid	No (Piecemeal resection)	Sheet of PD	None	ND
5	8	1	45/ female	Abdominal pain	ND	19.0 (EUS) 20.0( Pathologic analysis)	No	G1	Yes	Sheet of PD and BD	None	14
6	9	1	57/ male	None	Smooth surface	10.0 (EUS+ Pathologic analysis)	No	G1	ND	Sheet of PD	Postoperative pancreatitis	24
7	10	1	60/ male	None	Smooth surface with pannus	10.0 (Pathologic analysis)	No	G1	Yes	Sheet of PD and BD	None	12
8	11	1	42/ female	Abdominal pain	Smooth surface with pannus	16.0 (ND)	No	G1	Yes	ND	None	16
9	12	1	77/ female	None	Surface depression	≥20.0 (ND)	No	ND	ND	None	papillary Disinsertion/ perforation	ND
10	13	1	51/male	None	Surface depression with pannus	24.0 (Pathologic analysis)	No	Carcinoid	Yes	ND	None	18
11	14	1	57/ female	None	Surface depression	15.0 (ND)	No	ND	Yes	ND	None	14
12	15	1	63/male	Abdominal pain	ND	ND	ND	G1	ND	ND	None	ND

BD, bile duct; PD, pancreatic duct; EUS, endoscopic ultrasonography; ND, not described.

## Data Availability

The data presented in this study are available on request from the corresponding author.

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
