# Peer review of "Can Ampullary G1 and G2 Neuroendocrine Tumors Be Cured by Endoscopic Papillectomy?"

_jcm, 2023, doi:10.3390/jcm12062286_

Round 1

Reviewer 1 Report

The authors should improve the discussion. Firstl of all they should give recommendations when to operate when endoscopic treatment. Moreover they should some additional information about further supporting treatment, do the patients receive somatostatin analoga, was the tumor hormone active? Did they receive interferon alpha? What about molecular treatment: Sutinib, VEGF, PDGF And tyrosine kinase inhibitors?

Author Response

Dear professor,

Thank you very much for your constructive suggestions, which improved our manuscript significantly.

We have made the related revisions according to your suggestions as follows:

The authors should improve the discussion. Firstl of all they should give recommendations when to operate when endoscopic treatment. Moreover they should some additional information about further supporting treatment, do the patients receive somatostatin analoga, was the tumor hormone active? Did they receive interferon alpha? What about molecular treatment: Sutinib, VEGF, PDGF And tyrosine kinase inhibitors?

Answer: In Discussion section, paragraph 2, Line 4, we have added the statement “That is, endoscopic papillectomy could be performed for ampullary neuroendocrine tumors only when distant metastasis, regional lymph nodes metastasis and tumor in-vasion within the bile or pancreatic duct were eliminated, otherwise surgical tech-niques or other treatments should be considered”. Moreover, in Methods section, paragraph 1, Line 24, we have added the statement “Of note, no other further supporting treatments, such as interferon alpha or molecular treatment, were performed for these 7 patients”. Finally, we have improved the discussion section according to your instructions.

Reviewer 2 Report

This study ingestigated the feasibility of endoscopic intervention for duodenal neuroendocrine tumors. I have some suggestions listed below. 

1. How to recognize the tumor with G1/G2 characteristics rather than G3 before the definite resection?

2. As to the technical demand, is there any difference from the perspective of ampullectomy between the patients with ampullary tumor and NET?

Author Response

Dear professor,

Thank you very much for your constructive suggestions, which improved our manuscript significantly.

We have made the related revisions according to your suggestions as follows:

This study investigated the feasibility of endoscopic intervention for duodenal neuroendocrine tumors. I have some suggestions listed below.

  1. How to recognize the tumor with G1/G2 characteristics rather than G3 before the definite resection?

Answer: In this study, before the procedure, computerized tomography (CT), magnetic resonance imaging (MRI) and endoscopic ultrasonography (EUS) were performed to rule out the distant and regional lymph nodes metastasis; and endoscopic retrograde cholangiopancreatography (ERCP) was performed to ensure that there was no tumor invasion within the bile or pancreatic ducts. And the patients could underwent EP if no aforementioned advanced signs were found. And the results of these 7 patients turned out to be G1 or G2.

  1. As to the technical demand, is there any difference from the perspective of ampullectomy between the patients with ampullary tumor and NET?

Answer: The endoscopic treatment technique of ampullary NET was similar to that of other ampullary tumors. However, to date, the studies of endoscopic papillectomy for ampullary neuroendocrine tumors were very limited. Therefore, this study aimed to assess the feasibility of endoscopic papillectomy (EP) for ampullary neuroendocrine tumors.

Reviewer 3 Report

Journal of Clinical Medicine

Review of the manuscript Can ampullary neuroendocrine tumor (G1 or G2) be cured by endoscopic papillectomy.’

General comments

It is a well-written original paper. However, it could benefit during the process of minor revision.  Please apply the suggestions for the paper if you believe they will improve your document. Re-review is not required.

The suggestions are as follows:

1.     The title, Version 2: Can ampullary G1 and G2 neuroendocrine tumors be cured by endoscopic papillectomy. The title, Version 3: Can ampullary well and moderately differentiated neuroendocrine tumors be cured by endoscopic papillectomy.

2.     I don’t recommend using the abbreviations ANETs and EP, CT, EUS, and MRI in the main text of this manuscript. I recommend a good style: ampullary neuroendocrine tumor, endoscopic papillectomy, computed tomography, computed tomography scan, etc.  

3.     Be sure you use American English (for example, tumor) everywhere in the manuscript. For me, ‘tumour’ is an example of good writing (British English).

4.     Line 19: are there any references to endorse the second sentence?

5.     Line 23-24: the last part of the sentence is unclear (… and review the available literature sources).

6.     Line 28: you could say 'at this hospital’ (to avoid the abbreviation PLA). The alternative (line 27, to include the abbreviation PLAGH). 

7.     Line 30: I suggest including the Year (5th Edition).

8.     Lines 41, 46, and, I think, somewhere else: inserted (instead of implanted)

9.     Table 1, the 8th column: delete the abbreviation NET (leave only G1 and G2)

10.  Table 1, the footnote: ‘PD, pancreatic duct, BD, bile duct’ to add.

11.  Line 89: reconsider the structure of this sentence (you can divide this sentence into two sentences)

12.  Line 86: Galetti et al (not GALETTI)

13.  Table 2, 1st column: No. (not ‘no’)

14.  Table 2, the last column: ‘No recurrence (months)’, to bold it; please delete the word ‘period’.

15.  Table 2, the footnote: ‘PD, pancreatic duct, BD, bile duct’ to add.

16.  Line 103: I find ‘potions’ (should it be ‘the main options for patients with ampullary neuroendocrine tumors’?)

17.  Regarding the references. The journal recommends a different style. Please double-check it. If so, please precisely adjust references to the journal’s requirements.

The reviewer

18 February 2023

Author Response

Dear professor,

Thank you very much for your constructive suggestions, which improved our manuscript significantly.

We have made the related revisions according to your suggestions as follows:

  1. The title, Version 2: Can ampullary G1 and G2 neuroendocrine tumors be cured by endoscopic papillectomy. The title, Version 3: Can ampullary well and moderately differentiated neuroendocrine tumors be cured by endoscopic papillectomy.

Answer: We have changed the title to “Can ampullary G1 and G2 neuroendocrine tumors be cured by endoscopic papillectomy?” according to your instructions.

  1. I don’t recommend using the abbreviations ANETs and EP, CT, EUS, and MRI in the main text of this manuscript. I recommend a good style: ampullary neuroendocrine tumor, endoscopic papillectomy, computed tomography, computed tomography scan, etc.

Answer: We have deleted these abbreviations according to your instructions.

  1. Be sure you use American English (for example, tumor) everywhere in the manuscript. For me, ‘tumour’ is an example of good writing (British English).

Answer: We have tried our best to use American English everywhere in the manuscript.

  1. Line 19: are there any references to endorse the second sentence?

Answer: We have added the related references.

  1. Line 23-24: the last part of the sentence is unclear (… and review the available literature sources).

Answer: We have added the related references.

  1. Line 28: you could say 'at this hospital’ (to avoid the abbreviation PLA). The alternative (line 27, to include the abbreviation PLAGH).

Answer: We have changed this section to “at this hospital”.

  1. Line 30: I suggest including the Year (5th Edition).

Answer: We have added the Year.

  1. Lines 41, 46, and, I think, somewhere else: inserted (instead of implanted)

Answer: We have changed implanted to inserted in the manuscript.

  1. Table 1, the 8th column: delete the abbreviation NET (leave only G1 and G2)

Answer: We have deleted the abbreviation NET in Table 1.

  1. Table 1, the footnote: ‘PD, pancreatic duct, BD, bile duct’ to add.

Answer: We have added the footnote of PD and BD in Table 1.

  1. Line 89: reconsider the structure of this sentence (you can divide this sentence into two sentences)

Answer: We have divided this sentence into two sentences.

  1. Line 86: Galetti et al (not GALETTI)

Answer: We have changed GALETTI to Galetti.

  1. Table 2, 1st column: No. (not ‘no’)

Answer: We have changed no to No.

  1. Table 2, the last column: ‘No recurrence (months)’, to bold it; please delete the word ‘period’.

Answer: We have made the related revisions.

  1. Table 2, the footnote: ‘PD, pancreatic duct, BD, bile duct’ to add.

Answer: We have added the footnote of PD and BD in Table 2.

  1. Line 103: I find ‘potions’ (should it be ‘the main options for patients with ampullary neuroendocrine tumors’?)

Answer: We have changed potions to options.

  1. Regarding the references. The journal recommends a different style. Please double-check it. If so, please precisely adjust references to the journal’s requirements.

Answer: We have changed the references style to that of MDPI.

Round 2

Reviewer 2 Report

As to my previous question 1, How to define G staging rather than TNM staging in your study population? G staging depends on the histology rather than image study. 

Author Response

Dear professor,

Thank you very much for your constructive suggestions, which improved our manuscript significantly.

As to my previous question 1, How to define G staging rather than TNM staging in your study population? G staging depends on the histology rather than image study.

Answer: To be honest, it was very difficult to define G staging before the procedure. In this study, we found that G2 cases always showed a performance of surface depression, however, G1 cases tended to show a performance of smooth surface with pannus.

Thank your again for your efforts for our manuscript, feel free to contact me if there is any other problem.

Best regards,

Enqiang Linghu
